# Lysine Methyltransferase EhPKMT2 Is Involved in the In Vitro Virulence of *Entamoeba histolytica*

**DOI:** 10.3390/pathogens12030474

**Published:** 2023-03-17

**Authors:** Susana Munguía-Robledo, Esther Orozco, Guillermina García-Rivera, Jeni Bolaños, Jesús Valdés, Elisa Azuara-Licéaga, Mario Alberto Rodríguez

**Affiliations:** 1Center for Research and Advanced Studies of the IPN, Department of Infectomics and Molecular Pathogenesis, National Polytechnic Institute, Mexico City 07360, Mexico; 2Center for Research and Advanced Studies of the IPN, Department of Biochemistry, National Polytechnic Institute, Mexico City 07360, Mexico; 3Posgraduate Program in Genomic Sciences, University Autonomous of Mexico City, Mexico City 03100, Mexico

**Keywords:** *Entamoeba histolytica*, lysine methylation, lysine methyltransferases, stress response, in vitro virulence

## Abstract

Lysine methylation, a posttranslational modification catalyzed by protein lysine methyltransferases (PKMTs), is involved in epigenetics and several signaling pathways, including cell growth, cell migration and stress response, which in turn may participate in virulence of protozoa parasites. *Entamoeba histolytica*, the etiologic agent of human amebiasis, has four PKMTs (EhPKMT1 to EhPKMT4), but their role in parasite biology is unknown. Here, to obtain insight into the role of EhPKMT2, we analyzed its expression level and localization in trophozoites subjected to heat shock and during phagocytosis, two events that are related to amoeba virulence. Moreover, the effect of EhPKMT2 knockdown on those activities and on cell growth, migration and cytopathic effect was investigated. The results indicate that this enzyme participates in all these cellular events, suggesting that it could be a potential target for development of novel therapeutic strategies against amebiasis.

## 1. Introduction

Protein methylation is involved in stability, localization and interaction of cellular proteins with their binding partners [1]. This posttranslational modification (PTM) can be found on different amino acid residues, and it is known that methylation of arginine and lysine on histones is involved in epigenetic regulation of transcription [2]. Additionally, methylation of non-histone proteins has been shown to be important in regulating various cellular signaling pathways [3]. This PTM is catalyzed by protein methyltransferases that use S-adenosyl-methionine (SAM or AdoMet) as a methyl group donor and are classified according to their substrates.

Lysine methylation is mediated by protein lysine methyltransferases (PKMTs), which transfer up to three methyl groups (denoted as Kme1, Kme2 and Kme3) to the ε-amine of the side chain of lysine on protein substrates. The first characterized PKMTs were shown to methylate histones; thus, the study of these PTMs was largely focused on epigenetic regulation of transcription. However, nowadays, it is known that they can also modify non-histone proteins, regulating several cellular processes [3]. Most PKMTS contain a 130-amino-acid domain termed SET (referred to as Drosophila Su(var)3-9, Enhancer of zeste (E(z)) and Trithorax), which constitutes the active site [4]. Through systematic survey of the human genome for methyltransferases-related enzymes, 51 putative PKMTs were identified based on their content of an SET domain [5], although the activity of most of them has not been demonstrated yet. PKMTs have also been found in protozoa parasites, where it has been demonstrated that they are involved in epigenetics and motility, which in turn affect the virulence and/or stage transition of these parasites [6,7,8,9].

*Entamoeba histolytica* is the etiologic agent of human amebiasis, a disease that annually affects approximately 50 million people and causes up to 100,000 deaths worldwide [10]. Adherence to target cells, amoebapores- and proteases-mediated citotoxicity, phagocytosis and stress response are events necessary for its virulence and pathogenicity [11]; however, little is known about the molecules that regulate these activities. This microorganism has four PKMTs (EhPKMT1 to EhPKMT4), whose recombinant proteins showed the ability to transfer methyl groups to commercial histones [12]. Furthermore, antibodies against histone-methylated lysines showed that EhPKMT1, EhPKMT2 and EhPKMT4 catalyze epigenetic marks H4K20me2, H3K4me3 and H4K20me3, respectively [12], strongly suggesting that these PTMs occur in the pathogen and may participate in epigenetic regulation of transcription. On the other hand, during erythrophagocytosis, EhPKMT2 and EhPKMT4 were also detected in phagocytic cups and erythrocyte-containing vacuoles [12], proposing that these enzymes might somehow regulate this virulence-related property. However, until now, the role of EhPKMTs on the biology of *E. histolytica* remains unknown.

Here, to obtain insight into the role of EhPKMT2, we examined its expression level and localization in heat-shocked trophozoites and during erythrophagocytosis. In addition, we analyzed the effect of its knockdown on cell proliferation, viability of trophozoites under heat shock as well as on virulence in vitro. The results indicate that this enzyme participates in all these cellular events, suggesting that it could be a potential target for development of novel therapeutic strategies against amebiasis.

## 2. Materials and Methods

### 2.1. Entamoeba histolytica Cultures

Trophozoites of *E. histolytica* (strain HM1:IMSS) were axenically cultured in TYI-S-33 medium and harvested during the logarithmic growth phase as previously described [13].

### 2.2. Structural 3D Modeling

The 3D model of EhPKMT2 was obtained from the I-TASSER server (https://zhanglab.dcmb.med.umich.edu/I-TASSER/ (accessed on 17 January 2022) [14,15,16]. The RAMACHANDRAN PLOT server (https://swift.cmbi.umcn.nl/servers/html/ramaplot.html (accessed on 16 May 2022) was used to assess the quality of the most energetically stable model. The 3D structures of proteins were visualized and compared with the UCSF Chimera software (https://www.cgl.ucsf.edu; accessed on 6 March 2023).

### 2.3. Expression of EhPKMT2 in Trophozoites Exposed to Different Conditions

To evaluate the expression level of EhPKMT2 (EHI_069080) under heat shock, trophozoites were incubated at 42 °C for 15, 30 and 60 min. In addition, for erythrophagocytosis, trophozoites were incubated with fresh human red blood cells (RBCs) (ratio 1:25) for 10, 20 and 30 min at 37 °C. After the established times for each treatment, total extracts were obtained in the presence of proteases inhibitors (Complete Mini, Roche-Mannheim, Mannheim, Germany) and submitted to Western blot assays using the anti-EhPKMT2 antibody [12] (dilution 1: 500) and subsequently with a peroxidase-coupled secondary antibody (1: 10,000) (anti-rabbit IgG, GenTex, Carbondale, PA, USA, catalog no. GTX 213110-01). Antigen recognition was acquired by chemiluminescence (ECL Plus, GE-Healthcare, Chicago, IL, USA). As an internal loading control, membranes were exposed to an anti-beta actin antibody (1:3000, Santa Cruz, catalog no. SC-47778). For semiquantitative comparisons, bands recognized by the antibodies were analyzed by scanning densitometry using ImageJ software (https://imagej.nih.gov/; accessed on 6 March 2023) and the EhPKMT2 data were normalized to EhActin content. The relative expression of EhPKMT2 in trophozoites under basal conditions (time 0) was arbitrarily taken as the unit.

### 2.4. Immunofluorescence and Confocal Microscopy

To investigate localization of EhPKMT2, parasites were grown on coverslides and subsequently subjected to the treatments described above; then, trophozoites were fixed and permeabilized with cold methanol for 5 min. Non-specific binding sites were blocked with 10% FBS in PBS. Next, samples were incubated overnight at 4 °C with the anti-EhPKMT2 antibody (dilution 1:50) and later with an Alexa 488-conjugated secondary antibody (Invitrogen catalog. no. A-11008, Waltham, MA, USA) (1:400). Nuclei were counterstained with 4′,6-diamidino-2-Phenylindole (DAPI) and, finally, cells were observed through a confocal microscope (Carl Zeiss LSM 700, objective 40×, N.A. 1.3, laser 488 nm: 2.0%) and processed with ZEN 2009 Light Edition Software (Zeiss, Jena, Germany).

### 2.5. Knockdown of Ehpkmt2

The full-length *Ehpkmt2* gene was amplified by PCR using specific oligonucleotide primers (sense: 5´-ccccgcggccgc atg gat ttt tcg tta aaa tat c-3´; antisense: 5′-ccccagatct ata ctc aac ata ttc agt ata-3′, underlined sequences correspond to the *NotI* and *BglII* recognition sites in sense and anti-sense primers, respectively) and *Ehpkmt2/GST* construct [12] as template. PCR reaction was performed in a 50 µL volume reaction containing 0.2 mM of each primer, dNTPs 0.2 mM, MgCl_2_ 3 mM and 1 U of Taq polymerase (Invitrogen). Amplification cycles comprised: (i) 3 min of denaturing step at 94 °C; (ii) 35 cycles of 30 s of denaturing step at 94 °C, 1 min of annealing step at 55 °C and 2 min of elongation step at 72 °C and (iii) 10 min of elongation step at 72 °C. The amplified gene was digested with *NotI* (New England Biolabs, catalog no. R3189S, Ipswich, MA, USA) and *BglII* (New England Biolabs, catalog no. R0144S) and inserted (in antisense) into the pSA8 vector [17].

Transfection was carried out as described [18] using 5 μg/mL of G418 to select the transfected trophozoites. For higher knockdown, parasites were grown in the presence of 40 μg/mL of G418. To confirm the EhPKMT2 knockdown, trophozoites were submitted to Western blot assays, and relative expression of the protein, determined as described above, was compared with that of trophozoites transfected with the empty vector, whose data were arbitrarily taken as 1.

### 2.6. Effect of Ehkmt2 Knockdown on Heat Shock Response

Trophozoites transfected with the construct containing the *Ehpkmt2* gene in antisense (EhPKMT2-KD) or with the empty vector (mock) were submitted to heat shock, and their survival at different times was determined by trypan blue exclusion.

### 2.7. Effect of Ehpkmt2 Knockdown on In Vitro Virulence of E. histolytica

Cell migration of EhPKMT2-KD and mock trophozoites was performed as described [19], whereas their erythrophagocytosis rate and cytopathic effect on MDCK cells were determined as previously reported [20].

### 2.8. Statistical Analysis

Values of all assays were expressed as the mean ± standard error of three independent experiments by duplicate. Statistical analyses were carried out using GraphPad Prism V5.01 software by two-way ANOVA and Student’s *t*-test.

## 3. Results

### 3.1. EhPKMT2 Is Structurally Related to AKMT of T. gondii and Human SMYD1

It is already known that protein structure comparison provides support for their function [21]; therefore, to further predict the role of EhPKMT2, its protein structure was assembled using software I-TASSER. The predicted structure of this enzyme displayed the typical core of methyltransferases, consisting of an arrangement of twisted seven-stranded β-sheets forming two lobes, where the catalytic SET domain is located in the middle of the N-terminal lobe (Figure 1A). Comparison of the putative structure of EhPKMT2 with reported crystals of PKMTs from different organisms showed a closer relationship to the atypical PKMT (AKMT) of *Toxoplasma gondii* (PDB: 6FND) (Figure 1B) and to the mammalian SMYD1 (PDB: 3N71) (Figure 1C). Regarding complete protein EhPKMT2, it demonstrated 15.5% structural identity to AKMT (coverage 89%) and 11.64% identity to SMYD1 (coverage 83%) (Figure 1B,C), whereas the SET domain displayed 19.1% and 14.02% identity with those of AKMT and SMYD1, respectively (Figure 1B,C). AKMT is a methyltransferase that controls the motile behavior of the parasite [22], whereas SMYD1 catalyzes methylation of histones and non-histone proteins involved in cardiac and skeletal muscle morphogenesis [23]. Despite the identity being below 20%, which is considered significant [24], there are no other crystal structures closely related to EhPKMT2. Therefore, these results suggest that EhPKMT2 could be involved in cell motility of the parasite through methylation of cytoplasmic proteins, but it also may participate in epigenetics controlling other cellular events.

### 3.2. Changes in Expression and Localization of EhPKMT2 during Heat Shock and Erythrophagocytosis

Lysine methylation has been associated with virulence of some protozoa and fungi [7,8,22,25,26,27]. Thus, to investigate whether EhPKMT2 participates in the virulence of *E. histolytica*, we decided to analyze its expression and cellular localization in two events related to the virulence of this parasite: heat shock and phagocytosis [11,28].

Regarding heat shock, Western blot assays showed that the intensity of the band recognized by the antibody against EhPKMT2 was similar in extracts from trophozoites incubated at 37 °C (time 0) and those from parasites incubated at 42 °C for 15 min; however, at latter times of heat shock, therecognition of this band was apparently lower (Figure 2A). Densitometric data of the EhPKMT2 band at each time point, normalized to EhActin content, confirmed similar expression of EhPKMT2 at 37 °C and after 15 min of heat shock, but it decreased by approximately 30% after 30 and 60 min of incubation at 42 °C (Figure 2B). On the other hand, immunofluorescence assays on trophozoites cultured at 37 °C showed that the enzyme is located in dots throughout cytoplasm and around the nucleus, and, in various cells, it was found towards one pole of the trophozoites near the plasma membrane and concentrated in some pseudopods (Figure 3). After heat shock, most of the protein was relocated to the cell membrane and pseudopods. Interestingly, at 30 min of incubation, EhPKMT2 protein was not only around the nucleus but also within it (Figure 3). In addition, at 60 min of heat shock, the protein was almost absent in many trophozoites, but, in those that maintained expression, it was detected mainly around the parasites (Figure 3). Taken together, these results suggest that EhPKMT2 could participate in regulating both function of cytoplasmic proteins and gene expression during heat shock response.

On the other hand, the Western blot assays showed that the amount of EhPKMT2 gradually decreased during erythrophagocytosis (Figure 4A). Semiquantitative analysis revealed that protein level diminished by about 40, 45 and 60% after 10, 20 and 30 min of incubation with red blood cells relative to that detected in trophozoites without interaction with RBCs (time 0) (Figure 4B). As described above, under basal conditions, the protein was detected in cytoplasm and around the nucleus, but, when the trophozoites were incubated with the RBCs for 10 to 30 min, the protein relocated to the phagocytic cups (Figure 5, arrows); we also observed in certain parasites that the protein was also located around some erythrocytes that had already been ingested (Figure 5, arrowheads), suggesting that this methyltransferase interacts with the phagocytic machinery for phagocytosis of the target cells.

### 3.3. Ehpkmt2 Knockdown Modified Cellular Proliferation

To further understand participation of EhPKMT2 in parasite biology, we evaluated the effect of its knockdown on cell proliferation and in vitro virulence. To accomplish this, we cloned the full-length *Ehpkmt2* gene in antisense into the PSA8 vector [17]; this construct was transfected into *E. histolytica* trophozoites and expression of EhPKMT2 protein was analyzed by Western blot assays at different concentrations of the selecting agent (G418). We did not observe a significant decrease in the level of EhPKMT2 when the transfected trophozoites (EhPKMT2-KD) were cultured in the presence of 5, 10 or 20 µg/mL of G418, but, at 40 µg/mL, the recognition of this protein was markedly decreased (Figure 6A). Densitometry showed that, at this drug concentration, the expression of EhPKMT2 diminished by around 30% with respect to parasites transfected with the empty vector (mock) (Figure 6A,B). To guarantee that knockdown was specific, we also analyzed the expression of another lysine methyltransferase (EhPKMT4), which showed no change in its expression in EhPKMT2-KD (Figure 6A,B), indicating that the knockdown was exclusive to EhPKMT2.

Then, we evaluated the growth kinetics of transfected parasites in TYI-S-33 medium. Growth curves showed that cellular proliferation of EhPKMT2-KD increased approximately two-fold compared to mock parasites (Figure 7A). These results indicate that EhPKMT2 negatively regulates trophozoite proliferation under normal conditions.

### 3.4. Knockdown of EhPKMT2 Alter Different Virulence Factors

The results described above suggested that EhPKMT2 could also participate in heat shock response and phagocytosis, two events related to the pathogenic mechanism of *E. histolytica*. To corroborate that this methyltransferase is involved in parasite virulence, we investigated the efficiency of EhPKMT2-KD trophozoites to survive heat shock, engulf erythrocytes, generate a cytopathic effect on mammalian cells and migrate.

Regarding survival of trophozoites under heat shock, we observed that knocked down parasites were more resistant to incubation at 42 °C than mock population, particularly after 30 min, where the difference was statistically significant (*p* < 0.05) (Figure 7B), suggesting that downregulation of EhPKMT2 is necessary to support parasite survival during heat shock.

When expression of EhPKMT2 was reduced, phagocytosis rate showed a decrease of about 20% at 10 min of interaction with RBCs but recovered at 20 and 30 min (Figure 7C). Consistently, the ability of EhPKMT2-KD trophozoites to destroy MDCK monolayers was decreased approximately 90% relative to the cytopathic effect displayed by mock cells (Figure 7D). Surprisingly, cell migration of EhPKMT2-KD augmented approximately two-fold relative to control trophozoites (Figure 7E). All these results indicate that EhPKMT2 participates in different processes related to virulence of *E. histolytica*. This enzyme seems to be required in the early stages of phagocytosis for methylation of cytoplasmic proteins, while it could negatively or positively regulate expression and/or activity of proteins involved in the cytopathic effect and cell migration.

## 4. Discussion

Studies on lysine methylation have been concentrated mainly on histones modification and its effect on chromatin-state-dependent processes, such as transcription, DNA replication and DNA repair. However, many non-histone proteins have been identified as targets of lysine methylation. Indeed, nearly 3000 non-histone proteins from human cells exhibit lysine methylation at approximately 5000 unique sites [29], indicating that this PTM is involved in many cellular processes by enabling protein–protein and protein–nucleic acids interactions as well as regulating stability and subcellular localization of the target proteins [30].

It has also been shown that lysine methylation and, consequently, PKMTs of fungi- and protozoa-parasites play an important role in many cellular processes of these microorganisms, including virulence [7,8,22,25,26,27]. In blood stages of *Plasmodiun falciparum*, 570 lysine-methylated proteins were identified, indicating that this PTM is widespread in this pathogen [31]. Moreover, expression of some PfPKMTs, as well as the epigenetic marks catalyzed by them, show dynamic changes during the parasite asexual erythrocytic cycle, suggesting that these enzymes regulate stage conversions [6]. In addition, the PKMT named *PfSETvs* catalyzes trimethylation of H3K36, which represses expression of *var* genes; therefore, this enzyme is involved in virulence of *P. falciparum* [7]. In *Toxoplasma gondii*, the PKMT known as AKMT regulates cytoskeleton dynamics because, when it is absent, the microorganism remains largely immotile, compromising invasion and egress of the parasite from the host cell and, therefore, disrupting the lytic cycle of *T. gondii* [22]. Thus, due to their fundamental role in biology of protozoan parasites, PKMTs have been proposed as possible pharmacological targets for treatment of diseases caused by these microorganisms [8,9]. In fact, compounds BIX-01294 and TM2-115, which inhibit activity of PKMT G9a, have been shown to reduce H3K4me3 levels in *P. falciparum* and arrest parasite growth at all stages of the intraerythrocytic life cycle; furthermore, TM2-115 treatment of *P. berghei*-infected mice reduces parasitemia [32].

Only four PKMTs were detected in the unicellular parasite *E. histolytica*, whose recombinant proteins showed methyltransferase activity on commercial histones [12], suggesting that they may participate in epigenetics but may also regulate other cellular processes. Consistent with this hypothesis, two EhPKMTs, in addition to being found in the nucleus, were also detected in the cytoplasm around ingested erythrocytes during phagocytosis, suggesting that these enzymes participate in this virulence-related property [12]. Here, to gain insight into the role of EhPKMT2 in *E. histolytica* biology, we determined its putative 3D structure and expression level and subcellular localization in trophozoites subjected to heat shock and during erythrophagocytosis, two virulence-related properties [11,28]. We also analyzed the effect of its knockdown on the survival of trophozoites submitted to that stress condition and on virulence in vitro.

It was previously found that the SET domain of EhPKMT2 is phylogenetically related to the human SMYD family [12], whose members methylate a variety of histone and non-histone proteins that regulate several cell functions, including chromatin remodeling, transcription, signal transduction and cell cycle [33]. SMYD proteins, in addition to the SET domain, are characterized by the presence of a zinc finger motif named Myeloid-Nervy-DEAF1 (MYND) [33]. However, this motif is missing in the parasite protein.

We observed that knockdown of EhPKMT2 led to an increase in parasite proliferation; in accordance, in cell lines from epithelial ovary cancer, the augmented presence of H3K4me3 produces an antiproliferative effect [34]. Interestingly, recombinant PKMT2 produced H3K4me3 on commercial histones. The mammalian PKMTs that methylate H3K4 include members of the KMT2/MML and SMYD families as well as SET7/9 and PRDM9 [35]. Remarkably, the putative structure of EhPKMT2 is similar to the human SMYD1. On the other hand, SET7/9 also methylates β-catenin Lys180, YAP1 Lys499, STAT3 Lys140, E2F1 Lys181 and pRB Lys810, modifications that inhibit cell proliferation [36], and deletion of Set7/9 increases expression of cyclin A2 and D1, leading to accumulation of cells in S phase [37]. Therefore, we suggest that EhPKMT2, either by methylation of H3K4 or non-histone proteins, such as EhActin, EhTubulin or EhCyclins, has an important role in *E. histolytica* proliferation.

A proteomic profile of migratory neural crest cells identified 182 cytoplasmic lysine-methylated proteins, several of which are cytoskeleton-related [38], suggesting an essential role for this PTM in cytoskeletal proteins during neural crest migration. Moreover, overexpression of the PKMT known as SUV39H1 activates migration of breast and colorectal cancer cells [39]; SETD2 trimethylates α-tubulin lys40 to regulate mitosis and cytokinesis [40] and methylates β-actin lys68, participating in its polymerization [41]; EZH2 methylates actin-binding protein talin, a key regulatory molecule in cell migration, disrupting its binding to F-actin, leading to turnover of adhesion structures [42]. All these results indicate that PKMTs are involved in structure and dynamics of cytoskeleton. Here, we observed that EhPKMT2: (i) has a close structural relationship with the *T. gondii* AKMT and with the human SMYD1, which participate in parasite motility and muscle morphogenesis, respectively [22,23]; (ii) it is located in pseudopods and phagocytic cups and iii) its knockdown modified cell proliferation and migration, as well as phagocytosis, events that depend in part on the structure and dynamics of the cytoskeleton. These results suggest that EhPKMT2 could methylate cytoskeletal-related proteins, playing a significant role in its structure and dynamics.

Pathogenesis of *E. histolytica* relies on its capacity to adhere and lyse the colonic epithelium and subsequently spread to other organs, such as liver, lungs and brain [43]. This process is determined by the presence of many different proteins, such as EhAdhesins, EhCysteine-proteases and amoebapores [44], by its great phagocytic capability [28], by immune system evasion [45] and, recently, it has been shown that the heat shock response also participates in virulence [11].

PKMTs are involved in stress response through methylation of histone and non-histone proteins. In *C. elegans*, epigenetic mark H3K4me3 is required for the response to bacterial infection, xenotoxicity and heat shock [46], and trimethylation of Hspa8 lys561 improves its stability and function in chaperone-mediated autophagy [47]. We found that EhPKMT2 decreased its expression during heat shock, was relocated to inside the nucleus and cell membrane and its knockdown led to increased cell survival under this condition. Additionally, it has been described that, in *E. histolytica* during heat shock, expression of some cysteine-proteinases is disrupted while others increase [48]. These results suggest that EhPKMT2 is involved in the stress response and virulence by regulating gene expression and possibly altering stability of heat shock proteins of the parasite.

Cell migration is positively or negatively regulated by different PKMTs; knockdown of SMYD3 provokes loss of cell migration [49], whereas knockout of SET7/9 improves it [37]. We evaluated the effect of EhPKMT2 knockdown on migration of *E. histolytica* trophozoites because this process is required to invade different tissues after disruption of the intestinal epithelium [50]. Interestingly, diminishing of EhPKMT2 expression induced an increase in cell migration. Therefore, although this protein has a structural relationship to SMYD1, its function in both cell proliferation and cell migration is similar to that performed by SET7/9 [36,37].

Considering all the results, we propose that EhPMKT2 is associated to virulence in many ways. First, since histones are one of the best-described targets of PKMTs, it is possible that EhPKMT2 within the nucleus activates or inactivates gene expression of virulence-related proteins (Figure 8). On the other hand, it is possible that regulation of virulence also occurs directly on the proteins of cytoplasm. We suggest that cytoskeletal dynamics are regulated by EhPKMT2 through methylation of cytoskeletal proteins, such as EhActin or EhTubulin, which are involved in trophozoites migration (Figure 8). Furthermore, as EhPKMT2 accumulates on phagocytic cups, it could regulate activity of proteins present on these structures, such as EhAdhesins, EhRabs and EhESCRT proteins (Figure 8) [51,52,53]. On the other hand, this enzyme could regulate expression and/or stability of heat shock proteins, such as EhHSP70, which in turn modify expression of cysteine proteinases (EhCPs) during heat shock stress (Figure 8) [48], causing an increase in the cytopathic effect. In addition, EhPKMT2 could directly participate in regulating activity of proteins involved in this event, such as protein kinases (EhPKs) or calcium-binding proteins (EhCABPs) (Figure 8). Further studies on identification of PKMT2 targets, protein–protein interactions and in vitro assays are needed to discover the molecular mechanisms by which this methyltransferase regulates different cellular processes in *E. histolytica*. Nevertheless, although the client proteins for EhPKMT2 are unknown so far, this enzyme is involved in virulence, making it an attractive target for development of new therapeutic strategies against amebiasis.

## 5. Conclusions

The results indicate that EhPKMT2 downregulates survival of *E. histolytica* trophozoites under heat shock, but, by epigenetics and/or by controlling the activity of cytoplasmic proteins, this enzyme positively regulates the cytopathogenicity of the parasite. Therefore, this methyltransferase is an attractive target for development of new therapeutic strategies against amebiasis.

## Figures and Tables

**Figure 1 pathogens-12-00474-f001:**
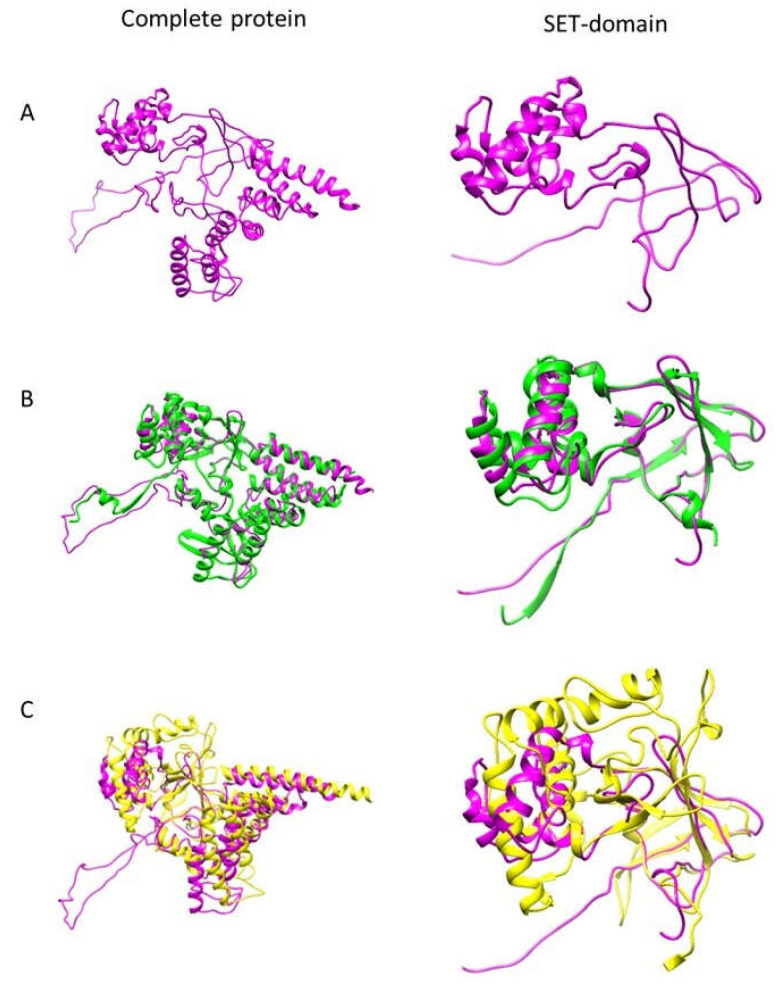
Structural modeling of EhPKMT2 and its comparison with AKMT and SMYD1. Software I-TASSER was used to predict the 3D structure of EhPKMT2, and then the whole protein and its SET domain were compared with the most closely related crystals (AKMT and SMYD1). (**A**) EhPKMT2 predicted structure. (**B**) Comparison of EhPKMT2 with AKMT (structural coverage 89%). (**C**) Comparison of EhPKMT2 with SMYD1 (structural coverage 83%).

**Figure 2 pathogens-12-00474-f002:**
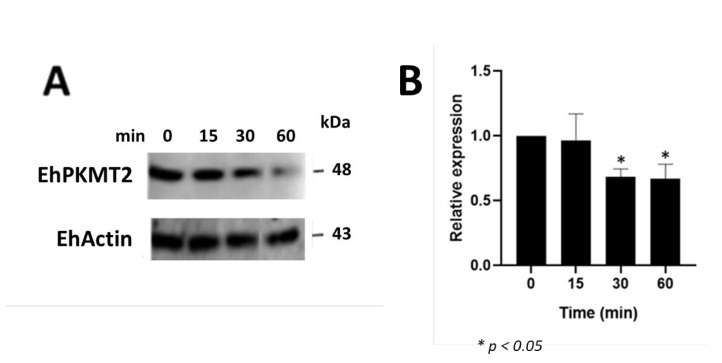
Expression of EhPKMT2 during heat shock. (**A**) Total extracts of trophozoites cultured at 37 °C (0 min) or incubated at 42 °C during 15, 30 and 60 min were submitted to Western blot using the α-EhPKMT2 antibody. As a loading control, same membranes were probed with an α-actin antibody. (**B**) The band detected by the α-EhPKMT2 antibody was analyzed by densitometry and the data were normalized to the EhActin content. The relative expression at time 0 was taken as 1. Data are expressed as the mean ± standard error of three independent experiments.

**Figure 3 pathogens-12-00474-f003:**
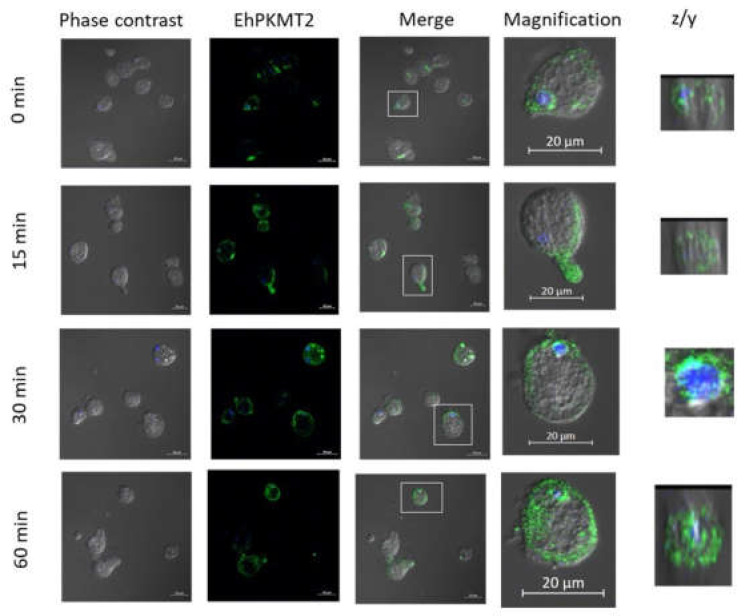
Localization of EhPKMT2 during heat shock. Confocal immunofluorescence assays on heat-shocked trophozoites using the α-EhPKMT2 antibody and subsequently an ALEXA 488-labeled secondary antibody (green). Nuclei were stained with DAPI (blue). Bars = 20 µm.

**Figure 4 pathogens-12-00474-f004:**
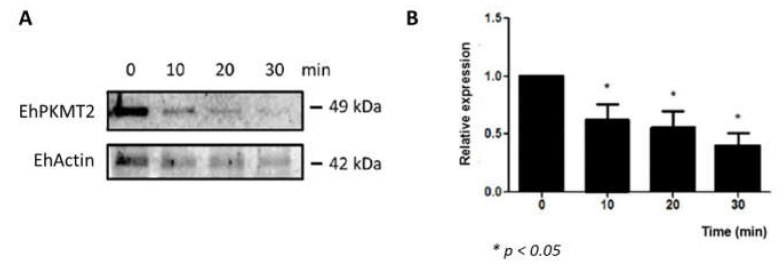
Expression of EhPKMT2 during phagocytosis. (**A**) Total extracts of trophozoites in the absence of human erythrocytes (0 min) or incubated at 37 °C with human erythrocytes for 10, 20 and 30 min were submitted to Western blot using the α-EhPKMT2 antibody. As a loading control, same membranes were probed with an α-actin antibody. (**B**) The band detected by the α-EhPKMT2 antibody was analyzed by densitometry and the data were normalized to the EhActin content. The relative expression at time 0 was taken as 1. Data are expressed as the mean ± standard error of three independent experiments.

**Figure 5 pathogens-12-00474-f005:**
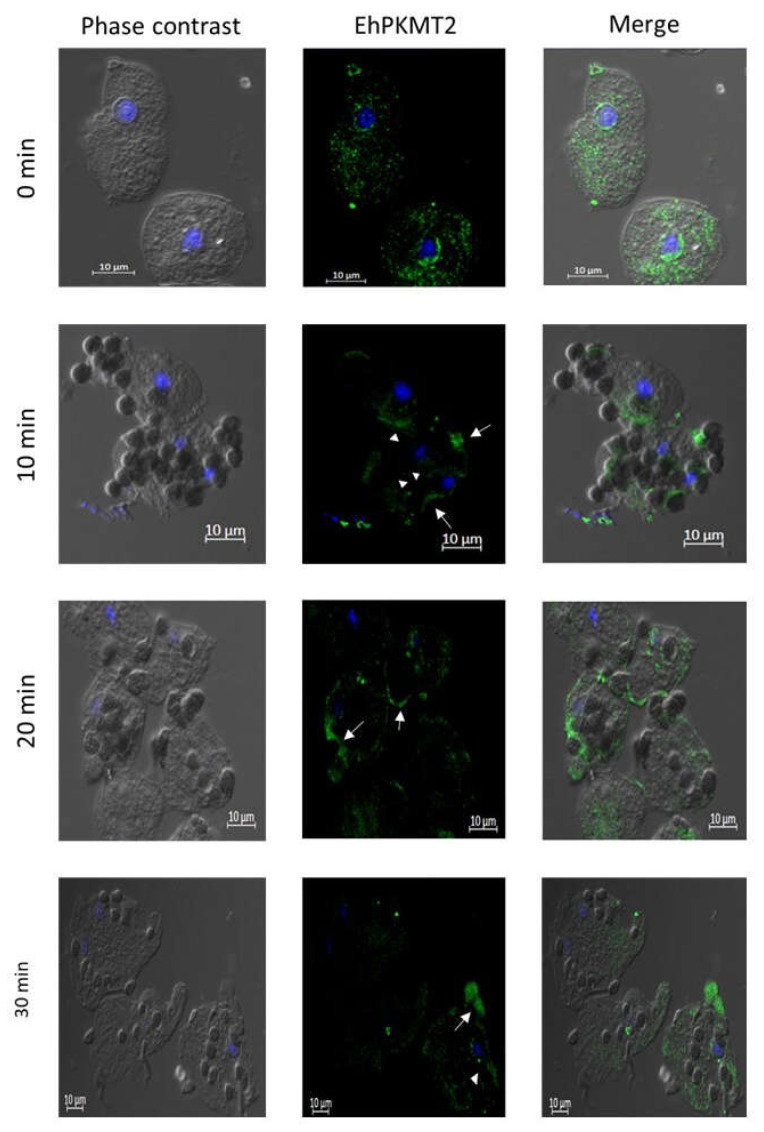
Localization of EhPKMT2 during erythrophagocytosis. Confocal immunofluorescence assays on trophozoites submitted to different times of erythrophagocytosis using the α-EhPKMT2 antibody and subsequently an ALEXA 488-labeled secondary antibody (green). Nuclei were stained with DAPI (blue). Arrows indicate phagocytic cups. Arrowheads indicate some ingested erythrocytes surrounded by EhPKMT2. Bars = 10 mm.

**Figure 6 pathogens-12-00474-f006:**
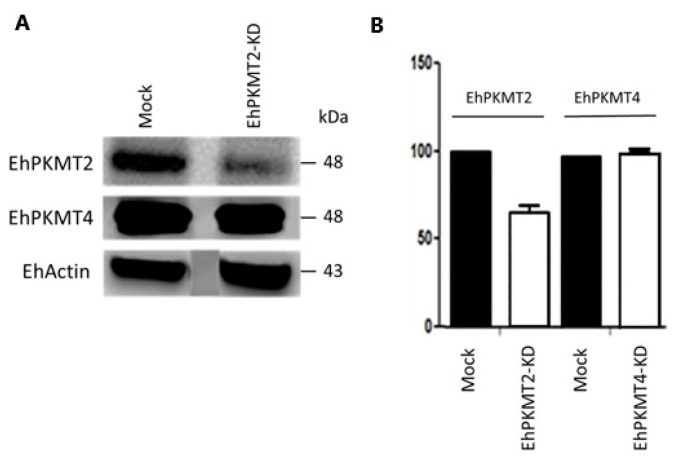
Knockdown of Ehpkmt2. The Ehpkmt2 gene was cloned in antisense into the pSA8 vector and trophozoites were transfected with this construct. Trophozoites transfected with the empty vector (mock) were used as a control. (**A**) Western blot on total extracts of EhPKMT2-KD and mock trophozoites using α-EhPKMT2, α-EhPKMT4 and α-actin antibodies. (**B**) The bands detected by the α-EhPKMT2 and α-EhPKMT4 antibodies were analyzed by densitometry and the data were normalized to the EhActin content. The relative expression in mock trophozoites was taken as 1. Data are expressed as the mean ± standard error of three independent experiments.

**Figure 7 pathogens-12-00474-f007:**
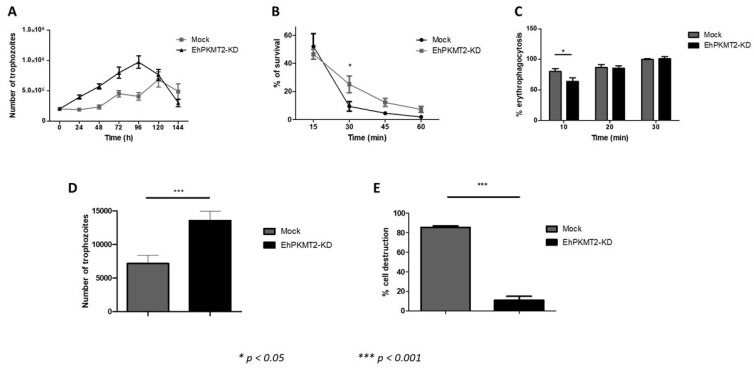
Effect of Ehpkmt2 knockdown on proliferation, heat shock and in vitro virulence. (**A**) Cell proliferation of EhPKMT2-KD and mock trophozoites. (**B**) Trophozoites were incubated at 42 °C and their viability was determined at 15, 30 and 60 min. (**C**) Trophozoites were incubated with human erythrocytes at different times. At each time, the non-phagocytosed RBCs were hypotonically lysed, and, after exhaustive washes, trophozoites were lysed and the hemoglobin released by ingested erythrocytes was determined at 400 nm. (**D**) MDCK cell monolayers were incubated with trophozoites (relation 1:1) at 37 °C during 2 h. Then, monolayer destruction was evaluated by methylene blue staining [20]. (**E**) Trophozoites were placed in the upper chamber of transwell inserts, whereas medium containing 10% of bovine adult serum was added to the lower chamber. After 3 h of incubation at 37 °C, the number of trophozoites that migrated to the lower chamber was counted. Data are expressed as the mean ± standard error of three independent experiments.

**Figure 8 pathogens-12-00474-f008:**
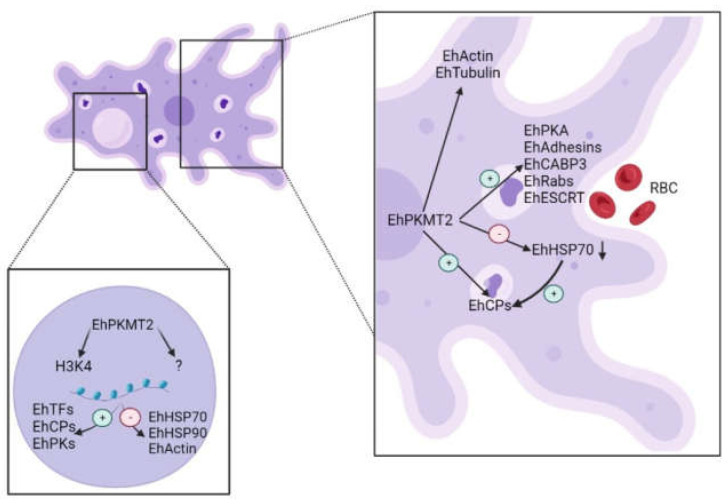
Model of the role of EhPKMT2 in *E. histolytica*. EhPKMT2 is involved in virulence, possibly regulating the activity of different proteins of *E. histlolytica* (Eh). +, activation; −, inactivation; H3K4, Lys4 of histone3; EhTFs, transcription factors; EhCPs, cysteine proteinases; EhPKs, protein kinases; EhHSP, heat shock proteins; EhPKA, protein kinase A; EhCABP3, calcium-binding protein 3; EhESCRT, endosomal sorting complexes required for transport; RBC, red blood cells.

## Data Availability

Data contained within the article are available upon request.

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
