# Peer review of "Lysine Methyltransferase EhPKMT2 Is Involved in the In Vitro Virulence of Entamoeba histolytica"

_pathogens, 2023, doi:10.3390/pathogens12030474_

Round 1

Reviewer 1 Report

In this paper Munguía-Robledo et al analyze the role of EhPKMT2 during heat stress and during phagoyctosis, they found that EhPKMT2 downregulates the survival of E. histolytica trophozoites under heat shock, contrarily, regulates the cytopathogenicity of the parasite.

Is a complete research, only they have to arrange the figure 8, in which the actin word is in spanish.

Reviewer 2 Report

Please provide a white background in Figure 1. It is very hard to make out anything. Keep the dimensions correct for individual panels. Current figure appears stretched. The authors mention structural identity. How was it determined? The authors should describe it in the methods. There are no details of the PDB structures used for the comparison. Authors need to clarify why they are considering structural identity of <20% as significant. 

For result section 3.2, authors determine localization differences post heat stress. They don't check the expression levels or localization patterns of any other protein except EhPKMT2 that might be affected by altering expression/localization of EhPKMT2. How do they reach to the conclusion that this enzyme regulates function of cytoplasmic proteins and gene expression during heat stress?

Figure 4A loading control is not equal. It seems the actin band of 0h is double the intensity than the 10 20 or 30 minute timepoints. I strongly recommend the authors to repeat this experiment. 

Round 2

Reviewer 2 Report

I am happy with the corrections made.

Author Response

Thank you four your feedback, the suggestion made by the reviewer were very important and they improved the quality of our paper